# Family Structure and Child Behavior Problems in Australia, the United Kingdom, and the United States

**DOI:** 10.3390/ijerph20031780

**Published:** 2023-01-18

**Authors:** Nina A. Stoddard-Bennett, Jordan Coburn, Mikaela J. Dufur, Jonathan A. Jarvis, Shana L. Pribesh

**Affiliations:** 1Department of Sociology, Brigham Young University, 2008 JFSB, Provo, UT 84602, USA; 2Department of Educational Foundations and Leadership, Old Dominion University, Norfolk, VA 23529, USA

**Keywords:** family structure, child mental health, internalizing behavior, externalizing behavior

## Abstract

A large body of literature suggests that children living with two married, biological parents on average have fewer behavior problems than those who do not. What is less clear is why this occurs. Competing theories suggest that resource deficiencies and parental selectivity play a part. We suggest that examining different contexts can help adjudicate among different theoretical explanations as to how family structure relates to child behavior problems. In this paper, we use data from the Growing Up in Australia: Longitudinal Study of Australian Children (LSAC), the UK Millennium Cohort Study (MCS), and the US Early Childhood Longitudinal Study (ECLS-K) to examine the relationship between family structure and child behavior problems. Specifically, we look at how living in several configurations of biological and social parents may relate to child behavior problems. Findings suggest both similarities and differences across the three settings, with explanations in the UK results favoring selectivity theories, US patterns suggesting that there is a unique quality to family structure that can explain outcomes, and the Australian results favoring resource theories.

## 1. Introduction

Family is the foundation for children that sets up how they are introduced to and interact with the rest of the world. A large body of research suggests that the family structures children grow up in influence children’s lives across a wide variety of outcomes such as in the educational, social, cognitive, and behavior realms [1]. However, changes in family structure can disrupt this process, which can result in greater instability and stress as well as fewer resources. These changes can in turn influence children’s mental health, socialization, and future success. For example, living in single-parent, stepparent, or cohabiting families is associated on average with lower academic achievement, including a lower high school GPA [2] as well as lower achievement test scores among both high school and elementary school children [3,4]. Children who live in homes with their biological parents who are married to each other on average enjoy better physical health than do their counterparts in single-parent or stepparent families [5,6]. Their mental and emotional health may be affected in similar ways; for example, Carballo et al. [7] find negative associations between living in family structures with access to both biological parents and both eating disorders and attention deficit hyperactivity disorder. These family structure associations seem to apply whether the biological parent is removed by marital or relationship dissolution [8], by death [9], by overseas military deployment [10], by immigration [11], or by incarceration [12]. These associations are also present for children parented by a mother who was never in a co-residential relationship with a partner—where a parent was not “removed”, but was never present [13]. Youth who live with neither biological parent on average also experience negative effects [14]. Thus, a large body of literature suggests that the number of biological parents available to children, and the formal relationship between those parents, can be an important factor in the family environments children experience and the outcomes they achieve [15,16,17].

In this paper, we examine family environments in different social contexts to evaluate competing theories striving to explain why family structures and transitions are linked to child health outcomes. In this case, we focus on child behavior problems. Behavior problems can be broken down into two categories: internalizing and externalizing. Children who exhibit internalized or overcontrolled behavior direct problematic energy towards themselves and are often overly anxious, clingy, feel inferior, or withdraw themselves in social situations as they are reluctant to engage with their peers or caretakers. In contrast, children who exhibit under-controlled or externalized behavior direct problematic energy towards others and often engage in actions such as bullying, lying, temper tantrums, or destroying toys [18]. Classifying behavior problems into these categories allows for more nuanced analysis since both internalizing and externalizing behaviors can have negative implications for mental health. Failure to address these behavior problems can lead to the development of mental illnesses such as anxiety and depression [19] or antisocial personality disorders and substance abuse [20].

In addition, such behavior problems are associated with worse academic and attainment outcomes. McLeod and Kaiser [21] find that behavior problems among six- to eight-year-old children have negative implications for secondary school completion and college attendance up to 14 years later. Both internalizing and externalizing behavior problems strongly diminish the probability of secondary school graduation, owing largely to academic failures. Among students who do complete secondary school, externalizing behavior problems reduce the probability of college enrollment, with the key mechanisms appearing to be the persisting effects of early behavior and academic predispositions. Analyzing the potential associations between changes in family structure and child behavior outcomes, as we do here, may also allow for more insight into how to promote healthy and pro-social child development in other areas, such as educational, social, and cognitive outcomes.

Studies have shown that living in family structures with two biological parents is on average associated with fewer child internalizing and externalizing behavior problems compared to single-parent families [22]. Similarly, some research suggests links between a broader set of family structures and other problematic behavior such as substance abuse or delinquency [15,23]. Research also finds that the relationship between family structure and behavior problems can be long-term [24]. However, these studies leave open the question of whether family structure itself affects behavior problems, or whether there are mechanisms associated with but distinct from family structures that are actually the key.

### 1.1. Resources

In other words, it is less clear why differences in family structures and exposure to transitions across such family structures might influence children’s lives. One potential explanation focuses on resource availability. This perspective argues that parents in some family structures are able to provide more physical or social resources that can facilitate child development than parents in other structures. It is not, under this model, the family structure that matters, but the way family structures might impinge on unmarried or stepparents’ ability to provide necessary resources [25]. For example, single parents do not have opportunities for dual incomes in their home, and single mothers, who make up the majority of single parents, are further hindered by gendered wage differences in the workforce.

In addition to differences in earned income, many stepparents have financial obligations to children born into previous relationships which may dilute their resources across additional children. In the presence of such resource dilution, children raised by only one biological parent may lack access to key resources that encourage positive development or to resources that protect against negative outcomes, such as access to health care. The situation may be even worse for children raised in settings where they have access to no biological parents, as supplementary income provided by the state may be insufficient to provide for all of their needs [26]. In an examination of the potential long-term effects of family structure on adult attainment outcomes, Lopoo and DeLeire [27] find that differences in adult income associated with childhood family structures are largely explained by the lack of resources available while living in those childhood families. Differences in income may be especially pronounced for single parents who may find part-time work necessary because of a lack of adult supervision [28]. In addition, some family structures are associated with a larger number of children in the home, which may be associated with resources being diluted across a greater number of family members [29].

Resource availability arguments, then, paint a convincing picture of possible mechanisms that would explain potential associations between family structures and transitions and child development, including the child outcomes we study here. However, while some research suggests that income differences help explain some of the gaps in attainment and behavior outcomes among children from different kinds of families, other studies find persistent effects of family structure when measures of resources that should close those gaps are included in models. Thus, resource availability alone may be insufficient to explain family structure effects.

### 1.2. Selectivity

Finally, some evidence suggests that the observed family structure or transition effects on child outcomes are likely reflections of selectivity processes. Parents are not randomly sorted into different configurations of partnering and childbearing, and as a result, children are not randomly sorted into different family configurations. Research finds that people who have children while cohabiting or prior to marriage are on average different on a number of demographic characteristics, including ethnicity, education, and labor market positioning, than people who delay childbearing until after marriage. For example, people who have children outside of formal relationships or while cohabitating on average have lower levels of education and are less well-placed in the labor market than people who have children within marriage [30]. Similarly, maternal age at birth, which is associated with a number of important child outcomes, is on average lower for mothers giving birth outside of marriage [30]. In the United States, minority ethnicity is also associated with non-martial childbearing, largely because of incarceration patterns targeted toward minorities [31]. Parents of children who end up living with no biological parents are especially likely to experience acute health or legal problems [29]. Proponents of the selectivity argument propose that the negative associations with being raised in non-traditional family structures can be explained by who these parents are rather than about the specific family structures they construct. For example, using models that examine within-child change, Amato and Anthony [9] found that while some effects of divorce on child outcomes persisted, differences existed across which children were most likely to be affected, with the largest effects for children whose parents were at the highest risk of divorce before separation happened. These parents may have fewer resources with which to mitigate typical family stressors. As a result, they may be less likely to enter marriages in the first place and less able to maintain successful relationships, deficits that likely spill over into their parenting. Negative effects on offspring according to this selectivity perspective, then, are less about structure or transitions and more about the parents making those decisions. Distinguishing among these theoretical approaches, however, continues to be difficult, at least in part because many data have been largely unable to make fine distinctions among family types [32].

### 1.3. Examining Additional Contexts: Cross-National Comparisons

One approach to distinguishing among theoretical explanations for the effects of family structure on child outcomes is to examine these associations cross-nationally. Little cross-national research has compared how the effects of living in different family structures may be similar or different across different social contexts and cultures. If differences across family structures are consistent across countries, this suggests that explanations are best sought within the interactions among family members. If the effects of family structure are significantly different across countries, this suggests that family structure differences are driven by access to physical or financial resources (i.e., differing social safety nets) or that different cultural approaches to families might reduce stressors sometimes associated with non-traditional family settings.

While cross-national comparisons therefore seem efficacious, such research on family structures, resources, and selectivity factors are limited. One study examining the effects of family structure on children’s perceived life satisfaction across 36 western countries found that while the family structure was associated with life satisfaction in bivariate models, strong welfare systems and lower economic gaps between rich and poor across the countries mediated perceived life satisfaction [33]. We do not yet know if similar patterns might hold for more proximate outcomes, such as children’s behavior problems. Research looking independently at Australia, the UK, and the US provide mixed findings about how similar the associations with family structure and behavior outcomes might be [34,35,36]. In these studies, researchers were often bound by data limitations and looked at two (intact vs. non-intact) or three (father-absent, mother-absent (both including stepparent families), and intact) family structure types in cross-sectional settings, an approach that masks important differences among family types [4,37]. These studies also employ data that, while of high quality, are somewhat dated and cannot consider shifts in normative attitudes towards families that have occurred in the past two decades. By contrast, a more recent study found that Australia’s welfare system arguably offers stronger buffers against poverty than those of the UK or US, which may mitigate the degree to which family structure or transitions influence child outcomes [22]. Comparing these countries directly and with newer datasets could give us a better understanding of any mechanisms that connect children in various family formations to behavior problems.

### 1.4. Research Questions

We investigate several possibilities here. One possibility is that potential negative associations between living in family structures that lack both biological parents can be explained by one of the theoretical arguments we have previously outlined. If this is the case, understanding the resources parents have access to in different family structures, the stressors children are exposed to in different family structures, or the personal characteristics of parents who create these families can illuminate why living in certain family environments is associated with poorer child outcomes. A competing possibility is that child internalizing and externalizing behavior problems will be strongly associated with exposure to families without both biological parents, and that these associations will persist even when accounting for other covariates. If this is the case, this would provide evidence that family structures themselves exert unique effects beyond the resources they provide, the stressors they incur, or the types of parents who create them. Finally, we compare results from these models across data from Australia, the United Kingdom, and the United States.

Given this information, our hypotheses are:

**H1a.** *Differences in behavior outcomes across family structures will be explained by differential access to resources*.

**H1b.** *Differences in behavior outcomes across family structures will be explained by differences in parental selectivity factors*.

**H1c.** *Differences in behavior outcomes across family structures will persist net of controls*.

We then examine any differences in evidence for these hypotheses across the three countries.

## 2. Methods

### Nations in the Analysis

All three countries used in our analyses are English-speaking countries that were selected for their similarities across important social and economic factors. Each of these high-income nations has a Gross Domestic Product (GDP) in the top 15 in the world [38]. In 2020, Australia had a per capita GDP of $51,692.8, the UK $41,124.5, and the US $63,413.5 [38] (Table 1). In addition, laws governing marriage and family formation in each of these countries derive from the common root of British law [39]. This includes similarities in how legal approaches concerning the disposition of property after family dissolution and definitions of partnership have developed in all three countries [40].

However, there are also distinct differences across these countries that provide interesting cross-cultural analyses. This includes differences in family structure patterns, maternal employment, prevalence of cohabitation, and government social safety nets. For example, the ages of first marriage for both men and women are similar in the UK and Australia but occur 2–3 years earlier in the US [41,42]. In addition, while the UK and US are similar in terms of their pre-COVID-19 pandemic unemployment rates, the rates in Australia are somewhat higher [43]. Additionally, the inequality of these nations varies, with the GINI score for Australia (32.5) suggesting a more egalitarian economic system compared to that of the UK and US (36.6 and 39.5, respectively) [44]. The differences in crude divorce rates show a higher divorce rate in the US (2.9) when compared to Australia and the UK (1.9 and 1.8, respectively) [45,46,47]. Similarly, there are sharp differences in the proportion of cohabitors across the three contexts, with cohabitation being much more common in the UK [48], and a much higher proportion in the US live in multigenerational households than the other two countries [49]. When looking at government social safety nets, both Australia and the UK offer more robust support for single parents than the US [50]. Though approaches to child support following martial dissolution originally grew from similar goals, applications of child support policy have diverged in the three countries [51]. This may present a variety of social, cultural, and economic differences between these countries that can provide more nuanced explanations of the impact that family structure may have on child behavior problems.

### 2.2. Data

To address critiques concerning the lack of longitudinal data used to examine the effects of family structures and to examine questions of the effects of family structures across multiple settings, we examine three longitudinal datasets looking at early childhood: one in Australia, the United Kingdom, and the United States.

#### 2.2.1. The Longitudinal Study of Australian Children: An Australian Government initiative (Australia)

The Longitudinal Study of Australian Children: An Australian Government initiative (LSAC) is a nationally representative sample of children born from 2003–2004. LSAC data were collected by the governmental Department of Social Services and the Australian Institute of Family Studies, and parental consent was obtained [52]. Data were gathered from Medicare Australia’s Medicare enrollment database, where children were ordered by date of birth and then a systematic sample was taken to recruit about 20 children per postcode. Recruitment for the main study took place between March and November of 2004, and the Australian data collection agency sought out children who were born between March 2003–February 2004, known as the “B [birth] cohort”. The initial sample was 5107 children. Again, we limited our data to children for whom we could identify measures of family structure and child behavior problem outcomes. We use data from the first five waves, which gives us a sample of 4085 children aged 8–9 years old. The LSAC collects data from parents, child caregivers, preschool teachers, schoolteachers, and as children age, from the children themselves. We recognize that while this sample is smaller than the other two samples from the UK and the US, it is still a large sample size with high-quality data and represents the population we wish to study.

#### 2.2.2. Millennium Cohort Study (United Kingdom)

The Millennium Cohort Study (MCS) follows a nationally representative sample of children born in the United Kingdom between 2000 and 2002. The Millennium Cohort Study was collected by The Centre for Longitudinal Studies at the University College London under the auspices of the National Health Service Research Ethics Committees, and both parents and children gave consent [53]. The initial survey targeted families who had a child aged nine months through governmental administrative records (see [54] for more information on the primary sampling frame). The initial sample was comprised of 18,818 children. We use data from the first four sweeps of data, when children are on average nine months, three years, five years, and seven years old, respectively. Retrospective questions in the nine-month data ask about circumstances at birth. Data are primarily gathered from a parent, though additional data from the target child are gathered as children age. We limited the sample to those children for whom we could identify reasonable measures of family structure and our outcome variables of measures of internalizing and externalizing behavior problems. This results in a sample of 14,920 seven-year-olds.

#### 2.2.3. Early Childhood Longitudinal Study-Kindergarten Cohort (United States)

The Early Childhood Longitudinal Study-Kindergarten Cohort (ECLS-K), collected by the National Center for Educational Statistics and the US Department of Education, draws a nationally representative sample of 21,260 children in the United States who were in kindergarten in the 1998–1999 school year and obtained parental consent [55]. The ECLS-K uses a multistage probability design, beginning with sampling schools and then sampling children of kindergarten age within schools (see [56] for more information on the ECLS-K sampling details). We use four waves of data, covering the fall of kindergarten, spring of kindergarten, first grade, and third grades, when children are around eight years old. As described in more detail in our measurement section below, we also use retrospective data provided by parents to determine the family structure at the child’s birth, creating an earlier “wave” of data for family structure. The ECLS-K collects data from parents, teachers, school administrators, assessments, and instruments given to target children and, in later waves, from the target children themselves. We limited our analytic sample here to children for whom we could identify reasonable measures of family structure and our outcome variables of measures of internalizing and externalizing behavior problems. This results in a third-grade sample of 16,151.

We note that although we are able to look at five time points, starting with birth, for all of the datasets, the time spans between data collection waves are not the same across the datasets. For example, the first sweep of data collection for the MCS and the LSAC concerns conditions at or near the target child’s birth (collected at nine months of age), while the second sweep of data is conducted two years later and every two years for the LSAC, but at three, five and seven years old for the MCS. By contrast, as described in more detail below, we use retrospective data to construct variables in the ECLS-K for birth, and the “next” wave of data is collected approximately five years later, as children enter kindergarten. Because we look at the possibility of effects of multiple potential changes over time on an outcome measured at the same age across all datasets, for most of our results here, this discrepancy is not an issue; however, we urge caution when looking, for example, at the change in proportions of children in family structures at different waves.

Once we determined these samples, missing data were of a generally trivial magnitude (for example, 196 missing cases on maternal age at birth in the ECLS-K). We imputed values for these cases via Stata 16′s chained multiple imputation protocol and creating 20 imputed datasets; examination of the trace plots of imputed means and standard deviations suggest appropriate random components introduced during imputation, and comparisons of observed, imputed, and completed data means and standard deviations increase confidence that the imputations created appropriate completed cases [57]. We follow Von Hippel [58] in excluding cases in which the dependent variable was imputed after the imputations were completed.

### 2.3. Measures

#### 2.3.1. Child Behavior Problems

In Table 2, we describe variables included in these analyses. The literature examining child behavior problems makes a distinction between internalizing behavior problems that focus on the self and externalizing behavior problems that focus on interaction with the social environment [59]. We measure child behavior problems with two commonly used factors as our dependent variables, one for internalizing behavior problems and one for externalizing behavior problems. To measure internalizing behavior problems, the LSAC used questions from five subsets of the Strengths and Difficulties Questionnaire (SDQ) [52]. Internalizing behaviors were taken from the SDQ subset about Emotional Symptoms [60]. These questionnaires were filled out by the teachers of the target child in Wave 5 when students were 8-to-9-years-old. In the MCS, we used eight questions from the Strengths and Difficulties Questionnaire that mapped well onto similar questions available in the ECLS-K, such as “child has many worries” and “child complains of headaches/illness/stomach aches” [56]. In the ECLS-K, we used data gathered from teachers (the Teacher SRS questionnaire) that asked about items such as anxiety, loneliness, low self-esteem, and sadness. To measure externalizing behavior problems, in the LSAC data, we used the SDQ questions about Hyperactivity and Conduct Scale [52]. For the MCS, we matched six SDQ variables that ask about the child’s behavior such as “child fights with others” and “child is restless, overactive, cannot stay still” [61]. In the ECLS-K, we used the same teacher data to measure the frequency with which a child argues, fights, gets angry, acts impulsively, disturbs ongoing activities, and talks during quiet study time [58]. Higher scores on each totaled scale indicate more behavior problems and worse child social adjustment.

We note that there are small but important differences in how the child’s behavior problems were measured in the original datasets. In the LSAC, variables were measured on a scale of 0–2. In the MCS, variables were measured on a scale of 1–3. In the ECLS-K, original variables that comprise the factor were measured on a scale of 1–4. These result in final behavior problems scales that have different ranges. To address this issue, we use standardized scores for all internalizing and externalizing behavior problem scales as our dependent variables. All results should be interpreted with this measurement strategy in mind, with coefficients associated with differences in standard deviations.

#### 2.3.2. Family Structure

We construct family structures primarily through parental reports of their relationship to the target child, their partnership status, and the relationship of any partner to the target child. Where necessary, we supplement these data with information from household rosters. This strategy allows us to consider the number of parents available to children in each wave of data collection, the relationship of the parental figure to the child (biological or social), and the marital status of partnered parental figures. We use the terms “stable” to represent families that have stayed constant since birth and “reconstituted” to represent families that have moved into other family structure categories, respectively. This results in categories of living with two stable married biological parents, two reconstituted married biological parents, two stable cohabiting biological parents, two reconstituted cohabiting biological parents, one biological and one social parent who are married to each other (reconstituted married non-bio), one biological and one social parent who are cohabiting (reconstituted cohabiting non-bio), one stable single biological parent, and one reconstituted single biological parent (single disrupted).

The small number of children who were living with no biological parents were spread too thinly across multiple alternate living structures to be examined in multivariate analyses; we exclude these cases from our sample. We applied this measurement strategy at age 0/1, 2/3, 4/5, 6/7, and 8/9 for the LSAC, at birth, age three, age five, and age seven for the MCS data, and at kindergarten fall, kindergarten spring, first grade, and third grade for the ECLS-K data.

The exception to the above strategy was using retrospective information provided by the responding parent to create family structures at birth for children in the ECLS-K. To do this, we used variables where the responding parent reported the year and month of marriage or cohabitation. If they were married/cohabiting before the sampled child was born, they were considered as married/cohabiting at the child’s birth. We acknowledge for both datasets that our measurement strategy does not account well for biological parents in a committed romantic relationship at the child’s birth who did not live together. The two-parent categories at birth include a very small number of families where the second parent is a social parent; these totaled fewer than 70 for all categories combined across all three datasets, too few to be analyzed as separate categories.

#### 2.3.3. Family Transitions

We then compared family structures across time to account for transitions that might introduce stressors to children’s lives. We account for transitions by constructing family structure variables for the 7–8-year-old data wave that considers both the family structure the target child lives in at that age and previous structures. This results in eight categories that combine elements of the number of parents available to children, the marital status of those parental figures, the biological or social nature of the relationship to the target child, and whether the structure the child lived in has changed since birth. We include categories for living in a two-biological-parent, stably married family; a two-biological-parent, stably cohabiting family; a stably single-parented family; a family where a disruption led to living with a single parent at the age of 7–8; a family where a transition led to living with both biological parents who were not married at the child’s birth but who later married; a family where a transition led to living with both biological parents who were neither married nor cohabiting at the child’s birth but who later moved into cohabitation; a family where a transition led to living with one biological parent and one social parent who are married at the time of the 7–8-year-old survey; and a family where a transition led to living with one biological parent and one social parent who are cohabiting at the time of the 7–8-year-old survey. Too few cases where a biological parent is married to or cohabiting with a social parent at birth exist to include them, or changes out of such structures, in the analyses. Two-biological-parent, stably married families are the comparison group in all analyses. We note that all the families who include coupled parents are opposite-sex couples; we identified only four families who appeared to include same-sex parents across the datasets, too few to produce reliable findings.

#### 2.3.4. Potential Explanatory Variables

To test theories concerning the lack of access to or dilution of important financial and physical resources in families where children do not have access to both biological parents, we include several measures of family resources. We measure resources first through a variable measuring family income. This is reported in US dollars in the LSAC and ECLS-K and British pounds in the MCS. In the MCS, income categories include 0£–3099£, 3100£ to 10,399£, 10,400£ to 20,799£, 20,800£ to 31,199£, 32,000£ to 51,999£, and 52,000£ and above. We recoded Australian and US data to comparable categories in their own currencies. We expect higher incomes to be associated with lower internalizing and externalizing behavior scores [62]. We also include measures of both paternal and maternal labor force participation. In addition to being associated with child development outcomes [63], Hakim [64] notes that female participation in the labor force differs across the US and the UK, with more women in the UK taking advantage of desirable part-time work opportunities that tend not to be available in the US. For the LSAC, parental employment is divided into nine categories: full-time work, part-time work, casual work, unemployed and seeking work, unemployed and not seeking work, full-time student, full-time home duties, permanently retired, and other. We note that the category for being out of the labor force conflates voluntary homemaking with involuntary unemployment; we would guess, based on US labor force trends, that this will primarily mask information about maternal labor choices. For the MCS, we construct a set of dummy variables for both parents that include full-time work, part-time work, being out of the labor force to care for children, out of the labor force for school, and out of the labor force but looking, other reasons for unemployment (such as poor health), and a category denoting there was no parent of that sex available to the child. Unfortunately, the ECLS-K does not include as much information on parental employment; the categories for the ECLS-K data are full-time work, part-time work, being out of the labor force, and no parent of that sex being available to the child. Full-time employment is the reference category. To measure resource dilution in these families, we include a variable measuring the number of children in the family in the 7–8-year-old wave, including the target child [32]. Stressors are measured by the aforementioned variables measuring transitions across family structures.

We test the possibility that potential differences in child behavior problems associated with the family structure might be due to the kinds of parents who form different types of family structures by accounting for several selectivity factors. We include a variable for maternal age at birth, a variable often associated with other family formation choices, which is measured in years [65]. We also include a set of variables to measure the highest level of parental education, which is often associated with family formation decisions [66]. We use the highest educational level reported by any parent living with the target child in the same wave of data as our child behavior problems variables. We include categories for less than high school education, high school completion, attending some college, completion of first college degree, and completion of high tertiary degrees. A-level certificates in the MCS data were included in the high school category to increase comparability to the LSAC and ECLS-K data. The MCS has one additional category, other educational qualifications; an examination of respondents who chose this option indicates that most have emigrated from other countries and likely held educational credentials that did not translate easily into the UK categories used by the MCS. High school/regular secondary school completion is the reference category. We also include a set of dummy variables to account for ethnicity, which is highly correlated with children’s family structure in the United States [67]. The LSAC data have no measures of ethnicity. In the MCS data, the categories are white, black—African origin, black—other origin, Indian, Pakistani, Bangladeshi, multiple ethnicities, and other ethnicities. In the ECLS-K data, the categories are white, black, Hispanic, Asian, and other ethnicity

#### 2.3.5. Additional Controls

We also account for variables often associated with child development and adjustment. We include a measure for the child’s age at the time of the behavior problems assessment since this impacts a child’s score [68]. This variable is measured in both week and months for LSAC, days for MCS, and months for ECLS-K. We also include variables for healthy birth circumstances: whether the target child experienced a preterm birth (1 = yes; 0 = no) and the child’s birth weight (measured in ounces) since both can impact development [69,70]. Finally, because girls often exhibit a faster mastery of social mastery skills [71], we include a variable measuring child sex (male = 1; female = 0).

### 2.4. Analytic Plan

We present two models for each dependent variable (one for internalizing behavior and one for externalizing behavior). Each of these models uses ordinary least squares regression to accommodate the distribution of the child behavior problems dependent variables. The first model includes only family structure variables to test whether exposure to family structures has any effect before we include measures of the main theoretical positions we outline above. The second model adds key resource and selectivity variables. We repeat these models for LSAC data, MCS data, and ECLS data. We note that it is not our purpose here to explain all variance in child behavior problem indicators; rather, we seek to see whether testing our models across the Australian, UK, and US contexts reveals similarities or differences in how resource availability theory or selectivity processes explain family structure and transition effects on child behavior problems.

## 3. Findings

### Family Structure

Figure 1 shows family structures in Australia, the UK, and the US when the children are ages 7–8. Children in Australia are more often in a biological married stable family structure (68.0%) than children in the United States (53.1%) and the United Kingdom (49.8%). Australia (7.6%) and the UK (7.2%) also have a higher percentage of children living in stable biological cohabiting families than does the US (1.1%). More children in the UK and US live in single stable biological family structures (8.7% and 7.2%, respectively) than in Australia (2.8%) and in reconstituted biological married families (11.1% in the UK and 7.2% in the US compared to 4.7% in Australia). However, reconstituted biological cohabitating families have similar prevalence rates across Australia, the US, and the UK data (1.0%, 2.7%, and 1.8%, respectively). In the US and Australia, reconstituted biological single-family structures are the second most common family structure (15.4% and 9.8%) but are the sixth most common among families in the UK (3.6%). Stepfamilies are also more common in the US (7.4%) than in Australia (1.5%) and the UK (2.2%). Finally, social families are much more common in the UK (14.7%) than in Australia (4.5%) and the US (3.4%). Taken together, stable biological married family structures are the most predominate family structure, and yet, there are important differences in the patterns of family structures in each country.

Table 3 and Table 4 present regression models predicting child internalizing and externalizing behavior problems for LSAC, MCS, and ECLS-K data. Model 1 for each dataset uses only the third-grade family structure categories as predictors. Model 2 adds family resources, selectivity variables, and additional controls. We note again that the dependent variable is a standardized behavior problems score to account for small measurement differences across the two datasets.

### 3.2. Internalizing Behavior Problems

#### 3.2.1. Australia

The Australian data in Model 1 shows that when compared to children with biological parents married since birth, children with single and stable, single and disrupted, and reconstituted cohabitating families (both biological and non-biological) report significantly more internalizing behavior problems, but this trend does not extend to children with biological cohabiting stable and married reconstituted families (both biological and non-biological). However, once the controls were added, only children in reconstituted biological cohabiting families were statistically different than children in biological married stable families (Model 2). As a result, Australian data provide substantially less evidence for H1c than either UK or US data do. Additional tests indicated that resources again had the biggest impact in accounting for differences, which provides the clearest evidence for H1a. When looking at resources, increased income was associated with decreased child behavior problems. Anything other than paternal full-time employment was associated with more internalizing behavior problems in Australia. Additionally, for maternal employment, part-time work was associated with an increase in behavior problems. In Australia, an increased number of siblings was associated with decreased internalized behavior problems.

Selectivity was not a significant factor when it came to assessing the relationship between family structures and child internalizing behavior problems in Australia. As a result, the Australian data provide no evidence for H1b. Higher parental education and maternal age at birth were not significantly associated with behavior problems. Recall that for the Australia data, information about race and ethnicity was not obtained. When looking at the other controls, a child’s age at assessment and maternal depression were positively associated with internalizing behavior problems, while birthweight was negatively associated with behavior problems; these factors, however, did not explain differences across family structures the way resources did, again pointing out that the Australian data are best explained by H1a.

#### 3.2.2. United Kingdom

Living in any family structure besides one having two stable biological married parents is associated with significantly more child internalizing behavior problems in the UK (Model 1). However, only children in single and stable, reconstituted and married non-biological, and reconstituted and cohabiting biological families remained significant once all the variables were introduced (Model 2). This indicates that while some family structures were still associated with worse child outcomes even after taking into effect resources, selectivity, and demographics, other differences across family structures can be explained by these theoretical mechanisms. For some family structures, then, there is evidence for H1c, where differences across family structures could be explained through other mechanisms. There is more evidence in the UK, however, for H1a, as supplemental analyses indicated that resources, including parental employment, income, and number of siblings, played an important factor in explaining differences in child behavior problems for several family structures. Homemaking by either parent was associated with more behavior problems, as were more siblings. Higher incomes were associated with lower behavior problems.

Selectivity also played a factor in the changes made from Model 1 to Model 2, but these effects were not as strong nor as numerous as those associated with resource explanations. Higher parental education was significantly associated with fewer internalizing behavior problems in the UK, as was greater maternal age at birth. When looking at race, there are important differences between the US and the UK in the differing distributions of non-white populations since ethnicity is measured in different ways. In the UK, being Black of African origin was associated with fewer internalizing behavior problems and being a member of an Asian minority group (aside from the Chinese group) or the “other” ethnic category had a significant, positive coefficient. Being a male child, preterm birth, and maternal depression were also associated with higher internalizing behavior problems in the UK. Findings from the UK provide some evidence for each of our three hypotheses, but more evidence for both H1b and, especially, H1a, than the US findings.

#### 3.2.3. United States

In Model 1, for internalizing child behavior problems in the US, living in any family structure besides one with two biological parents who have been married since before the child’s birth is associated with significantly more behavior problems. The significant differences in internalizing behavior problems between children with stable married biological parents and other groups persist for US children when variables accounting for resources, selectivity, and other controls are introduced except for children in biological cohabiting stable families, which are no longer significantly associated with higher behavior problems (Model 2). While the coefficients for most of the family structure comparisons remain statistically significant for the US when other variables are introduced to the model, the size of those coefficients decreases substantially, for some groups by as much as half. Supplemental analyses demonstrate that variables measuring family resources account for most of this difference. This is particularly true for income, where a higher income is significantly associated with lower child internalized behavior problems. For paternal employment, “other,” which is most likely associated with paternal homemaking or long-term unemployment, is associated with more internalizing behavior problems. The ways in which resources help to explain associations between family structure and child internalizing behavior problems provide evidence for H1a.

Selectivity variables also helped explain differences in internalizing behavior problems between American children living with stably married biological parents and those who were not. For parental education, only children with parents who have a higher tertiary degree saw a significant decrease in internalizing behavior problems. Race yielded an interesting finding. In the US data, all children from non-white groups (Hispanic, Black, and Asian) have significantly lower standardized internalizing behavior problems scores than do white children. While this initially seemed odd given the prevalence of negative outcomes associated with minority status in the US, we speculate that this may be related to teacher stereotypes concerning the ways minority children might display social adjustment problems. Additional controls found that being male, preterm birth, and maternal depression were all significantly associated with higher internalizing behavior problems scores in the US. The child’s age was negatively associated with behavior problems. Taken together, while resources were an important factor in accounting for some of the differences, data in the US remained resilient to many other controls. As such, the US data provide evidence primarily for H1c that there are unique mechanisms in different family structures that cannot be explained by other factors.

#### 3.2.4. Cross-Country Comparisons

Looking across all three countries, there is evidence in support of both hypotheses H1a and H1c. Resources had strong associations with child internalizing behavior problems in all three contexts and explained differences across some family structures in the US and UK and almost all family structures in Australia. Our findings across country contexts therefore provide strong support for H1a concerning the idea that differences in family structures are more due to the resources available to those families than the family structures themselves. However, there is also some support for H1c, or the idea that some family structures include unique mechanisms internally, since even when accounting for these covariates, there are some significant differences that cannot be explained. There is evidence for hypothesis H1c across all three countries, though the strongest evidence is produced by the US and UK data.

There are also some distinct differences across the countries that are instructive. Australian data were the most sensitive to the addition of controls, perhaps due to the LSAC’s small sample size (N = 3802) when comparing them to the US and UK (N = 16,158, N = 15,540; respectively). These findings could also be indicative of different services or governmental programs in Australia that help to alleviate any unique problems associated with family structures. While the findings in the US and UK are somewhat more similar, the differences in racial distributions and findings provide opportunities for additional exploration, especially around questions of selectivity.

### 3.3. Externalizing Behavior Problems

#### 3.3.1. Australia

Turning to externalizing behavior problems in Table 4, we find that similar patterns exist for the association between family structure and child externalizing behavior problems as those we found for internalizing behavior problems. Upon examining Model 1 for the LSAC data, most family structures do still reflect higher child externalizing behavior problems compared to children in stable married biological families, and neither children in biological cohabiting families nor reconstituted married biological families show any significant differences in externalized behavior problems compared to the reference group. This difference may reflect smaller sample sizes in the Australian data. After accounting for controls, children in single stable families were also no longer significantly associated with problem behavior. This is important because this is the second largest family structure in the Australian data, after stably married biological families, so it is not as easy to dismiss this result as a statistical artifact. These findings provide some support for H1c, which states that differences in behavior outcomes across family structures will persist net of controls. However, family structures are operating differently in Australia, with children in biological cohabiting families and reconstituted married biological families doing just as well as children biological married stable families. Once controls are accounted for, children in single stable families can also be added to that category.

Supplemental analyses for the Australian data also tell a story that diverges from the US and UK contexts. The data for siblings reflect that in Australia, each additional sibling is associated with about a 0.05 standard deviation *decrease* in child externalizing behavior problems. This is a very interesting finding, as more siblings was associated with higher externalized behavior problems in the other two countries and income was significant in lowering child externalizing behavior problems. In Australia, higher parental education is associated with fewer child externalizing behavior problems. Additionally, a child’s sex was not significant, and being older was associated with higher externalizing behavior problems. Birthweight is significant in Australia, with bigger babies having fewer externalizing behavior issues. Higher rates of maternal depression were significantly related to higher externalizing behavior problems. H1b, our hypothesis concerning selectivity factors, thus receives its strongest support when examining externalizing behavior problems among Australian children. However, in Australia, resources seem to be operating differently.

#### 3.3.2. United Kingdom

In the UK Model 1, looking only at family structure, living in any structure other than with stable married biological parents is associated with significantly more child externalizing behavior problems. For internalizing child behavior problems, only children in single stable families, reconstituted married biological families, and reconstituted cohabiting biological families remained significantly higher than two-married parent biological families. However, when looking at child externalizing behaviors in the presence of variables accounting for resources, selectivity, and other controls, all other family structures were significantly higher than the reference group of biological married stable families (Model 2). These data provide support for hypothesis H1c, which states that differences in behavior outcomes across family structures will persist net of controls. Though the coefficients were smaller in Model 2, we propose that there are characteristics or mechanisms of family structures that lead to differing mental health outcomes for children. Controls in all the theoretical blocks were similar in their effects as they were in the models reported above for internalizing behavior problems, but they were not able to fully explain differences across family structures.

#### 3.3.3. United States

For US data in Model 1, looking only at family structure, living in any structure other than with stable married biological parents is associated with significantly more behavior problems. Differences between children living with stably married biological parents and those in several other family structures remain significant and positive for US data in Model 2; however, two family structures are no longer statistically different from the stably married biological families: biological cohabiting stable families and reconstituted cohabiting biological families. Because all other family structures were resistant to control variables, this provides evidence for hypothesis H1c, which states that differences in behavior outcomes across family structures will persist net of controls. However, supplemental analyses show that resource variables explain the differences in externalizing behavior problems for children in biological cohabiting stable families and reconstituted cohabiting biological families, which provides evidence for the resources explanations in H1a. Additionally, it is important to note that while there continued to be increased externalized behavior problems for children in single and stable, single and disrupted, reconstituted and married biological, reconstituted and married non-biological, and reconstituted and non-married non-biological families, the coefficients for the increases of externalized behavior problems for children within these family structures were much lower than the coefficients seen in Model 1. This again provides evidence for H1a concerning resources. Control variables, including the key resources variables, operated very similarly as in the models reported above for internalizing behavior problems.

There is somewhat weaker evidence for H1b concerning selectivity processes in the US. Like models for internalizing behavior problems, higher parental education was associated with fewer child externalizing behavior problems. A child being female and older was associated with fewer externalizing behavior problems in the US. In the US, all non-white groups, except for the “other” group, have higher externalizing behavior scores than do white children; this is in contrast to internalizing behavior models, where their scores were lower. This may be due to the ways teachers expect minority children to demonstrate social adjustment problems. Preterm birth and maternal depression were associated with more child externalizing behavior problems in the US. These findings provide some support for H1b, which posits that mental health outcomes across family structures will be explained by differences in parental selectivity factors, but the fact that these selectivity variables did not fully explain significant differences across family structures means that evidence is weak by comparison to the evidence supporting H1a and H1c.

#### 3.3.4. Cross-Country Comparisons

By examining these results through the lens of the theories tested in this paper, we find that while coefficients for most family structure categories remain statistically significant when controls are added, providing the strongest support for H1c, selectivity variables alone are more helpful in reducing the size of family structure coefficients in the UK (MCS data), while resource and selectivity variables considered together are more important in the US (ECLS-K data). In Australia (LSAC data), only selectivity factors are important in explaining the family structure’s associations with child externalizing behavior problems resources.

These findings concerning child externalizing behavior problems provide support for all three hypotheses. Hypothesis H1a, which predicted that differences across family structures would be explained by differential access to resources, receives its strongest support from the US data. Additionally, there is evidence for H1b, which predicts that mental health outcomes will differ across family structure because of differences in parental selectivity factors, but most strongly in the Australian context. Finally, H1c predicts that differences in behavior outcomes across family structures will persist net of controls, suggesting internal mechanisms within family structures. We find that across all three countries, after accounting for all controls, there are still significant differences across many family structures; however, the family structures affected are different in each country. This suggests that with child externalizing behaviors, there are some family characteristics in certain family structures that tend to lead towards more externalizing behavior problems in children. It is most notable that in the UK, children in every family showed more behavior problems than the reference group of children in biological married families. In the US, these differences diminished for biological cohabiting families. However, in Australia, the differences diminished for single stable families. This is notable because, on average, single-parent families would have lower incomes and more time at work, leaving less time for their children and leading potentially to more behavior problems [72]. This is not the case in Australia and calls for a deeper examination of how social safety nets are helping these families.

## 4. Discussion

In this paper, we set out to compare how different family structure configurations and patterns in Australia, the UK, and the US affect child behavior problems, and whether family structure influences were better explained by resource, selectivity, or alternative theories. We found evidence for all three theories, which varied in results across internalizing and externalizing outcomes and the three countries. These mixed results add nuance to the complicated effects of diverging patterns of family formation and government support for families across these otherwise similar contexts [50]. Children who lived in stable cohabiting structures with both biological parents did not differ from those who lived with stable married biological parents after accounting for resources and selectivity issues in the US and Australia. Taken together, these findings suggest that family structure and stability effects may be tied to resources and selectivity processes that encourage the involvement of biological parents in the United States and Australia. Additionally, models using data from Australia and the United Kingdom were much more sensitive to covariates, especially when looking at child internalizing behavior problems. While this supports some policy initiatives concerning families, it also calls into question the extent to which those policy initiatives need to be centered around marriage rather than stable access to both biological parents. They also call into question the degree to which widely accepted policy approaches focusing on marriage as a panacea for child behavior problems are useful, given that the introduction of a married stepparent was still associated with lower child social adjustment, even in the presence of controls. Our findings suggest that policies may also need to consider the potential lasting impact of divorce on child outcomes even for remarried parents. At the same time, the fact that these patterns were not the same in the US data, and that resource and selectivity variables were not as useful in reducing family structure effects as in the Australia and UK data, indicates that looking at family structure and transition processes in different contexts can provide important insights into how family processes might shape child outcomes. Our findings are suggestive that stronger social safety nets in the form of cash transfers in Australia and the UK indicate an opportunity for interventions to strengthen parental health and other family characteristics that could benefit child health. The relative lack of such patterns in the US data may indicate additional and persistent needs to address poverty and other physical resource deficits in alternative family structures.

This suggests that lack of resources, lack of parental characteristics associated with successful family lives, and certain selectivity factors may be common challenges to families across different settings. However, policymakers should be sensitive to the ways opportunities to improve child health may vary across contexts. For example, when looking at Australia, some of these factors, such as income, the number of siblings, and parent education, operate in a way that equalizes more family structures [62,63,64]. It is important to note, however, that there were some differences across the countries in how resources and selectivity were associated with child social adjustment. Similarly, interventions in the UK might be best targeted for increasing access to parental education because low parental education was associated with lower child behavior problems only in the UK. On the other hand, differences in maternal work patterns across the countries suggest that the common patterns of part-time work among mothers in the UK could have beneficial effects for child health in the US and Australia if economic conditions that encourage such work existed in those countries [53]. Obvious racial differences existed across the US and UK as well. These findings indicate a need to delve more into sociocultural differences in potential covariates of family structures. We anticipate, for example, that differences in access to social services that might be more available in Australia and the UK might mitigate the effects of reduced family resources.

Of course, it is important to note how these mechanisms and resources are not things that can be “controlled away” in actual children’s lives. Selectivity arguments often focus on the “types” of parents who are likely to be lone parents, but do little to address the structural issues that shape their opportunities to do any other kind of parenting [56] Similarly, “controlling for” resources may help to explain differences in child social adjustment in the US for children who have access to both biological parents, but it does little to incentivize those parents to live together. These findings can help illuminate what might be useful in helping children from different family backgrounds adjust, but micro-level interventions are an inefficient way to deal with macro-level, institutional inequalities that are driving these patterns [73]. Future research, then, should focus on macro-level resources such as the social services we mention above.

In addition, this study has some limitations. For example, while looking at sibling/half-sibling presence and grandparent presence in the home was beyond the scope of this inquiry, these are increasingly common family configurations that bring additional complexity and could allow for more detailed examinations of how family members bring resources to the family or dilute them across additional family members [68]. Similarly, these data do not allow for the inclusion of half-siblings or social siblings who do not reside in the same household as the respondent but who could be contributing to resource dilution. Because of the nature of the outcomes we study here, we do not fully exploit the longitudinal nature of these data; looking at outcomes that vary over time could allow more sensitive tests on whether the timing of family transition stressors is a more important explanation of negative child outcomes [74]. Examining countries that differ more significantly in language, colonial history, and cultural practices than do the US, UK, and Australia could also provide finer tests of both macro-level and micro-level explanations. Additionally, the US data were collected when the children were in kindergarten, so the conditions at birth were imputed retrospectively and could lead to possible recall or attrition bias. We also acknowledge the possibility that the effects of factors such as resources and demographic characteristics could potentially differ across family structures. While future research could explore this possibility more thoroughly, we note that in other analyses using these data, we have not found moderating effects [75].

While these findings add to the mixed picture on the effects of family structure and transition [33], they also join other work in emphasizing the importance of considering carefully how different groups might react to family structure influences. Exciting work such as that looking at how children with different genetic profiles react differently to parental entrance and exit [76] highlights the ways children’s shifting environments are not destiny. Continued research examining more nuanced family structures and changes over time can help to illuminate the degree to which families trump—or do not—other factors influencing children’s lives.

## Figures and Tables

**Figure 1 ijerph-20-01780-f001:**
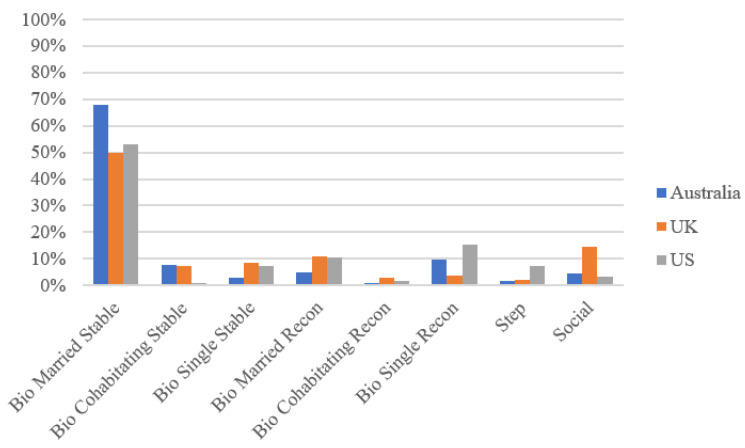
Family Structures in Australia, the United Kingdom, and the United States.

**Table 1 ijerph-20-01780-t001:** National Indicators for Australia, the US, and the UK.

Country	GDP Rank (2020)	GPD per Capita (2020)	First Age of Marriage (M/F)	Unemployment Rate (2019)	GINI (Year)	Crude Divorce Rate (per 1000)
Australia	13	51,692.8	32.2/30.06	5.2%	32.5 (2018)	1.9 (2020)
UK	5	41,124.5	33.4/31.5	3.74%	36.6 (2019)	1.8 (2016)
US	1	63,413.5	30.4/28.6	3.67%	39.5 (2019)	2.9 (2018)

Notes: GDP per capita in USD. GINI index from the most recent available year. US and Australia report median age at first marriage, UK reports mean age at first marriage.

**Table 2 ijerph-20-01780-t002:** Descriptive Statistics for All Variables in Analysis.

Variable		Proportion or *M*		Range
	Australia	UK	US	Australia, UK, US
Family Structure				(1, 8)
Biological Married Stable	0.68	0.50	0.60	
Biological Cohabiting Stable	0.08	0.07	0.02	
Biological Single Stable	0.03	0.09	0.07	
Post-Birth Biological Married	0.05	0.11	0.09	
Post-Birth Stepfamily	0.02	0.03	0.055	
Post-Birth Biological Cohabiting	0.01	0.04	0.02	
Post-Birth Social Family	0.05	0.02	0.03	
Post-Birth Transition to Single	0.10	0.15	0.11	
Internalizing Behavior	−0.01	0.00	0.00	(−1.06, 4.69) (−1.01, 5.68) (−1.11, 3.68)
Externalizing Behavior	−0.02	0.00	0.00	(−1.30, 4.56) (−1.28, 3.77) (−1.00, 5.70)
Father’s Employment				(0, 1) (1, 4) (1, 5)
Full-time	0.89	0.64	0.71	
Part-time	0.11	0.06	0.03	
Not in paid labor force		0.08	0.03	
Looking for work			0.02	
No partner to be employed		0.23	0.21	
Mother’s Employment				(1, 3) (1, 4) (1, 5)
Full-time	0.33	0.15	0.48	
Part-time	0.42	0.45	0.23	
Looking for work			0.03	
Not in paid labor force	0.25	0.40	0.24	
No partner to be employed		0.00	0.02	
Income (reported in quintiles)				(1, 5)
Bottom	0.24	0.19	0.34	
Second	0.20	0.19	0.13	
Third	0.20	0.20	0.30	
Fourth	0.18	0.21	0.11	
Top	0.18	0.21	0.12	
Number of Siblings	1.59	1.48	1.47	(0, 4) (0, 13) (0, 11)
Parent Highest Education	2.82			(1–5)
Less than secondary school		0.15	0.08	
Secondary school		0.48	0.22	
Some college		0.13	0.34	
Bachelor’s degree		0.16	0.19	
Graduate School		0.08	0.17	
Mother’s Age at Birth	31.2	28.52	28.57	(15, 48) (14, 51) (12, 46)
Child Race				(N/A) (1, 11) (1, 5)
White		0.85	0.59	
Black/Other Black		0.00	0.13	
Hispanic			0.18	
Asian/Other Asian		0.00	0.05	
Mixed		0.03		
Indian		0.02		
Pakistani		0.04		
Bangladeshi		0.01		
Black Caribbean		0.01		
Black African		0.02		
Other		0.00	0.05	
Child Gender				
Male	0.51	0.51	0.51	(0, 1)
Female	0.48	0.49	0.49	
Child’s Birth weight	121.2	118.39	118.46	(19.40, 191.89) (14, 255) (16, 219)
Pre-term Birth				(0, 1)
Not Pre-term	0.94	0.92	0.93	
Pre-term	0.06	0.08	0.07	
Child’s Age (years)	8.96	7.23	9.17	(8.17, 9.85) (6.34, 8.15) (7.92, 11.12)
Maternal Depression	7.08	4.48	1.35	(0.90, 7.94) (1–5) (1–4)

**Table 3 ijerph-20-01780-t003:** Ordinary Least Squares Regression of Internalizing Behavior on Family Structure, Resources, Selectivity and Other Controls in Australia, the United Kingdom, and the United States (N = 3802; N = 15,540; N = 16,158).

	Australia		UK		US	
	Model 1	Model 2	Model 1	Model 2	Model 1	Model 2
Family Structure						
Biological Cohabiting Stable	0.039 (0.061)	−0.041 (0.059)	0.119 *** (0.012)	0.065 *** (0.012)	0.270 *** (0.076)	0.089 (0.074)
Biological Single Stable	0.415 *** (0.096)	0.129 (0.095)	0.204 *** (0.012)	0.070 ** (0.021)	0.551 *** (0.030)	0.286 *** (0.077)
Post-Birth Biological Married	0.337 *** (0.054)	0.157 * (0.055)	0.144 *** (0.009)	0.041 * (0.019)	0.378 *** (0.022)	0.218 ** (0.071)
Post-Birth Stepfamily	0.003 (0.076)	−0.107 (0.073)	0.071 *** (0.010)	0.026 * (0.010)	0.202 *** (0.026)	0.119 *** (0.026)
Post-Birth Biological Cohabiting	0.505 *** (0.131)	0.462 *** (0.126)	0.172 *** (0.020)	0.088 *** (0.019)	0.451 *** (0.030)	0.323 *** (0.031)
Post Birth Social Family	0.737 *** (0.158)	0.524 ** (0.152)	0.187 *** (0.017)	0.082 *** (0.017)	0.218 *** (0.058)	0.070 (0.058)
Post-Birth Transition to Single	0.388 *** (0.077)	0.235 ** (0.076)	0.215 *** (0.021)	0.112 *** (0.021)	0.484 *** (0.043)	0.297 *** (0.044)
Resources						
Paternal labor force participation						
Part-time		0.122 * (0.061)		−0.011 (0.010)		−0.001 (0.043)
Homemaker				0.021 * (0.009)		
Student						
Unemployed, looking						0.059 (0.063)
Other						0.118 ** (0.044)
No father				−0.012 (0.014)		−0.058 (0.073)
Maternal labor force participation						
Part-time		0.177 *** (0.042)		0.001 (0.006)		−0.022 (0.020)
Homemaker				0.037 *** (0.007)		
Student						
Unemployed, looking						0.037 (0.045)
Other		−0.043 (0.036)				0.018 (0.020)
No mother				−0.062 (0.041)		0.041 (0.089)
Family income		−0.000 * (0.036)		−0.015 *** (0.003)		−0.030 *** (0.004)
Number of siblings		−0.040 * (0.017)		0.005 * (0.002)		0.006 (0.007)
Selectivity Issues						
Parental education						
High school or less		−0.024 (0.013)				
Some college				−0.042 *** (0.007)		−0.005 (0.033)
First degree				−0.053 *** (0.009)		−0.015 (0.033)
Higher tertiary degree				−0.055 *** (0.009)		−0.060 (0.037)
Other educational qualification				−0.059 *** (0.010)		−0.106 ** (0.039)
Maternal age at birth		−0.005 (0.003)		−0.002 *** (0.000)		0.002 (0.001)
Child race						
Hispanic						−0.120 *** (0.023)
Black				0.087 (0.045)		−0.150 *** (0.027)
Black—African origin				−0.038 * (0.016)		
Black—Other origin				−0.011 (0.019)		
Asian				0.068 * (0.016)		0.264 *** (0.036)
Chinese				0.040 (0.062)		
Indian				0.037 ** (0.014)		
Pakistani				0.086 *** (0.011)		
Bangladeshi				0.079 *** (0.019)		
Multiple ethnicities				0.013 (0.013)		
Other ethnicity				0.085 ** (0.032)		0.011 (0.036)
Other controls						
Child is male		0.041(0.030)		0.021 *** (0.004)		0.080 *** (0.016)
Child birthweight		−0.002 * (0.001)		−0.000 (0.000)		−0.000 (0.000)
Preterm birth		−0.060(0.071)		0.019 * (0.009)		0.098 ** (0.034)
Child age at assessment		0.118 * (0.050)		−0.009 (0.008)		−0.004 * (0.002)
Depression		0.288 *** (0.016)		0.095 *** (0.003)		0.007 *** (0.001)
Constant	−0.087 ***	1.58 **	1.244 ***	1.908 ***	−0.133 ***	0.53 ***

Note: Model 1 includes Family Structure (FS), Model 2 includes FS and Resources, Selectivity, and Other Variables (all variables). Comparison Groups: Married Stable, Female, Less than secondary school, Full-time work. * *p* < 0.05. ** *p* < 0.01. *** *p* < 0.001.

**Table 4 ijerph-20-01780-t004:** Ordinary Least Squares Regression of Externalizing Behavior on Family Structure, Resources, Selectivity, and Other Controls in Australia, the United Kingdom, and the United States (N = 3802; N = 15,540; N = 16,158).

	Australia		UK		US	
	Model 1	Model 2	Model 1	Model 2	Model 1	Model 2
Family Structure						
Biological Cohabiting Stable	0.039 (0.061)	−0.041 (0.059)	0.119 *** (0.012)	0.065 *** (0.012)	0.270 *** (0.076)	0.089 (0.074)
Biological Single Stable	0.415 *** (0.096)	0.129 (0.095)	0.204 *** (0.012)	0.070 ** (0.021)	0.551 *** (0.030)	0.286 *** (0.077)
Post-Birth Biological Married	0.337 *** (0.054)	0.157 * (0.055)	0.144 *** (0.009)	0.041 * (0.019)	0.378 *** (0.022)	0.218 ** (0.071)
Post-Birth Stepfamily	0.003 (0.076)	−0.107 (0.073)	0.071 *** (0.010)	0.026 * (0.010)	0.202 *** (0.026)	0.119 *** (0.026)
Post-Birth Biological Cohabiting	0.505 *** (0.131)	0.462 *** (0.126)	0.172 *** (0.020)	0.088 *** (0.019)	0.451 *** (0.030)	0.323 *** (0.031)
Post Birth Social Family	0.737 *** (0.158)	0.524 ** (0.152)	0.187 *** (0.017)	0.082 *** (0.017)	0.218 *** (0.058)	0.070 (0.058)
Post-Birth Transition to Single	0.388 *** (0.077)	0.235 ** (0.076)	0.215 *** (0.021)	0.112 *** (0.021)	0.484 *** (0.043)	0.297 *** (0.044)
Resources						
Paternal labor force participation						
Part-time		0.026 (0.055)		−0.005 (0.014)		0.046 (0.042)
Homemaker				0.036 ** (0.0013)		
Student						
Unemployed, looking						−0.042 (0.061)
Other		−0.026 (0.055)				0.108 * (0.043)
No father				−0.011 (0.019)		−0.057 (0.070)
Maternal labor force participation						
Part-time		−0.041 (0.036)		−0.026 ** (0.009)		−0.145 *** (0.020)
Homemaker				0.009 (0.010)		
Student						
Unemployed, looking						0.002 (0.043)
Other		0.064 (0.042)				−0.124 *** (0.020)
No mother				−0.047 (0.058)		−0.038 (0.086)
Family income		−0.000 (0.000)		−0.021 *** (0.004)		−0.022 *** (0.003)
Number of siblings		−0.049 ** (0.017)		−0.000 (0.003)		−0.044 *** (0.007)
Selectivity Issues						
Parental education		−0.063 *** (0.013)				
High school or less				−0.065 *** (0.009)		−0.033 (0.031)
Some college				−0.081 *** (0.012)		0.002 (0.032)
First degree				0.142 *** (0.012)		−0.098 ** (0.036)
Higher tertiary degree				0.126 *** (0.015)		−0.121 *** (0.037)
Other educational qualification						
Maternal age at birth		−0.006 (0.003)		−0.004 *** (0.001)		−0.001 (0.001)
Child race						
Hispanic						−0.103 *** (0.022)
Black				−0.073 (0.064)		0.193 *** (0.026)
Black-- African origin				−0.075 ** (0.023)		
Black-- Other origin				−0.005 (0.027)		
Asian				−0.011 (0.041)		−0.302 *** (0.034)
Chinese				−0.039 (0.087)		
Indian				0.023 (0.020)		
Pakistani				0.021 (0.016)		
Bangladeshi				−0.032 (0.026)		
Multiple ethnicities				−0.003 (0.018)		
Other ethnicity				−0.016 (0.046)		−0.013 (0.034)
Other controls						
Child is male		0.462 *** (0.031)		0.146 *** (0.006)		0.418 *** (0.015)
Child birthweight		−0.002 (0.001)		−0.001 *** (0.000)		−0.001 (0.000)
Preterm birth		−0.108 (0.071)		0.009(0.013)		−0.016 (0.033)
Child age at assessment		−0.007 (0.050)		−0.064 *** (0.011)		−0.006 *** (0.002)
Depression		0.165 *** (0.016)		0.093 *** (0.005)		0.006 *** (0.001)
Constant	−0.097 ***	1.580 **	1.43 ***	2.666 ***	−0.176 ***	0.748 ***

Note: Model 1 includes Family Structure (FS), Model 2 includes FS and Resources, Selectivity, and Other Variables (all variables). Comparison Groups: Married Stable, Female, Less than secondary school, Full-time work. * *p* < 0.05. ** *p* < 0.01. ***. *p* < 0.001.

## Data Availability

LSAC data may be acquired through contract with the Australian Data Archive (URL (accessed 17 January 2023): https://growingupinaustralia.gov.au/data-and-documentation/accessing-lsac-data). MCS data may be acquired through contract with the UK Data Service (URL (accessed 17 January 2023): https://beta.ukdataservice.ac.uk/datacatalogue/series/series?id=2000031). ECLS-K data may be acquired through contract with the National Center for Education Statistics (URL (accessed 17 January 2023): https://nces.ed.gov/ecls/kindergarten2011.asp).

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
