# Peer review of "Family Structure and Child Behavior Problems in Australia, the United Kingdom, and the United States"

_ijerph, 2023, doi:10.3390/ijerph20031780_

Round 1

Reviewer 1 Report

The main concern I have with this paper is that there is little motivation for the use of a multi-country comparison. What do we learn from this comparison beyond the outcome that many results are similar but some differ? While the countries are similar in many respects, there are likely important policy, institutional and social differences that could either generate more specific testable hypotheses or at least help to illuminate possible reasons for differences in results. For example, are there different legal and/or social treatment of cohabitating couples compared to married couples such as eligibility for benefits, or rules on child custody or dissolution of assets in the event of the relationship ending? Discussion in the paper (e.g. lines 687-689) only hints at the presence of such differences, but there is a general absence of context on any of the three countries included in the paper.

In 2.2 the paper notes that child behavioral problems in the LSAC were measured on a 0-2 scale while the others were on a 1-3 scale and 1-4 scale respectively. I understand that these were standardized in order to facilitate comparison but it is not clear what is reported for these variables in Table 1. If they are raw averages then how can two of the three numbers be 0 or negative? If they are standardized then why are there such pronounced differences across countries in the average values?

The authors are fond of the word ‘tapping’ to indicate measuring an effect. This is not conventional terminology.

In the Australian data there appears to be either an error or a marked difference in definition of full time work for fathers since only 11% are classified as full-time compared to 64% and 71% for the other countries. It doesn’t appear to be definitional differences since the % of mothers working full time shows a more plausible pattern across countries.

This point leads to another concern - the paper needs a more detailed discussion of limitations. Some are noted in the paragraph beginning ‘Future research…’ on p19 but there is little mention of limitations arising from differences in the comparability of the data. To give one example, in the US data, conditions at birth must be imputed using retrospective data collected when the children in the sample were in kindergarten. This raises the possibility of both recall and attrition bias. More generally, what were attrition rates like for each of the surveys over consecutive waves of data collection?

On line 683 the paper notes that marriage alone may not be a panacea for behavioral issues since families involving married step-parents still have worse outcomes. It may well be that both marriage and divorce are both very important factors, since the dissolution of a marriage of both parents is a major event in children’s lives.

The focus of the paper is on variables measuring different family structures and transitions, with an array of potentially explanatory variables included to measure financial and physical resources, among other factors. This approach basically treats these sets of variables as independent determinants of behavior outcomes (though there will be multicollinearity). What it does not allow for is the effect of a family transition on behavior to be modified by those other variables. It may well be for example that the impact of a marriage dissolution on child outcomes is moderated for families with greater economic means. This could be explored more through interactions of the key explanatory variables of interest

Reviewer 2 Report

Thank you for inviting me to review this manuscript. The study is interesting and the work is potentially important in understanding the effect of family structure on young children’s development. However, the biggest concern is that there lack a strong theoretical background or statement for testing these competing models, which could be added to substantiate your study. Other concerns are detailed below that I hope are helpful to the authors to further develop their manuscript.

1. Literature review: The subtitles are confusing: “1.1. Resources, Stressors, and Selectivity” “1.2. Selectivity”. In subsection 1.1, the authors seem to only review literature relevant to “resources”. Relevant theoretical frameworks are preferred to explain these three factors: resources, stressors, and selectivity. Then in subsection 1.3, the authors only mentioned one of the factors: “While cross-national comparisons therefore seem efficacious, such research on family 150 structures and resources is limited.” What about the other two factors?

2. In the potential explanatory variables, the authors indicated several variables included in this study, which should also be discussed in the literature review as to how the existing literature supported your choice of the explanatory variables.

3. Analytic plan. The details of data analysis should be presented. Including the use of statistical method, fit indices, etc. The authors separate the dependent variables (internalizing and externalizing behaviors) into two models. Since the two behaviors are closely related, the authors should try to see if putting the two variables into one model would affect your findings.

4. Findings. In 3.1 Family Structure, the results are descriptive, so it is not appropriate to use “more likely” unless differential analysis is performed. 

Round 2

Reviewer 1 Report

Thanks to the authors for the revised paper. I appreciate the effort that has been put in to addressing my comments, in particular my first point about motivating the use of a cross-country comparison. Since this paper's primary contribution is the cross country comparison, there needs to be clear demonstration about what can be learned from this comparison. However, the points of comparison listed and discussed in the paper omit those dimensions that would be most relevant to the research questions at hand, namely those directly relevant to family structure and changes therein. For example, are there any differences (or were there any differences over the period) in the legal recognition of civil unions? In the eligibility for spousal benefits? In divorce laws (e.g., at-fault vs. no-fault divorce, custody, division of assets, financial support, etc.)? In support for single parents? As well, the authors note that there are differences in government social safety nets across countries but they don't articulate any.

Reviewer 2 Report

 Thank you for the authors' effort in developing their manuscript. This revision looks fine to me.
